# Convergence or Divergence among Business Models of Public Bus Transport Authorities across the Globe: A Fuzzy Approach

**Büşra Buran** [1,*] and **Mehmet Erçek** [2]

1   Graduate School, Istanbul Technical University, Maslak, Istanbul 34469, Turkey
2   Management Engineering Department, Istanbul Technical University, Beşiktaş, Istanbul 34367, Turkey; ercekme@itu.edu.tr
*   Correspondence: buran18@itu.edu.tr

**Abstract:** Building on the debate about global convergence or divergence of practices, this study aims to query the viability of a new strategic action tool specifically geared to the interests of public bus transportation authorities (PBTA) around the globe and explore the degree of homogeneity in their responses as well as the possible drivers of them. To answer its research question, the study first offers a generic business model design for a PBTA, which integrates an extended version of the business model canvas with external environmental factors in order to enhance its sustainability. Subsequently, the importance attributions of international transportation experts to different model components are evaluated by using the Spherical Fuzzy AHP method. The model is developed in three hierarchical layers and evaluated by experts from four continents: America, Asia, Australia, and Europe. The results indicate that the expert opinions tend to converge more on the internal components of the model and diverge on the external components, especially regarding economic and technological factors. A strategic response action set is also designed to facilitate the adoption of the model by PBTA. The study not only extends the research on the strategic management of the public bus transportation domain but also contributes to the convergence and divergence debate by offering a reconciliatory duality perspective.

**Keywords:** public bus transportation authorities (PBTA); business model; Spherical Fuzzy Analytic Hierarchy Process (SF-AHP); convergence; divergence

## 1. Introduction

### 1.1. Aim of the Study

Despite a heavy hit from the COVID-19 pandemic, public transportation systems still represent the backbone of moving goods and people within metropolitan cities, as they combine marine, rail, and bus modes under their jurisdiction [1–9]. In populated metropolitan areas and especially in developing countries, bus transportation remains the central mode of transportation to reach relatively diverse and remote destinations within cities. As distancing measures have become prevalent after the pandemic, which strongly incentivize rapid digitalization across industries, public bus transportation authorities/agencies (PBTA) face additional challenges to cope with these abrupt transformations [10]. It is argued that the increasing digitalization spurred by advancements in the information and communication technologies (ICT) not only shape homogenizing forces at the demand side such as convergence of global consumer consumption patterns across countries [11] but also engender technological convergence at the supply side via fusion of diverse technological domains and applications [12]. Even though convergence of individual and firm-level practices indicates a long debate, within which many contend that convergence will not occur en masse and rapidly, there is growing evidence about the diffusion of convergent practices due to the intensification of technology. Thus, citizens of metropolitan areas around the globe increasingly witness shared mobility services, IoT-enabled smart communication platforms, or open application programming interface designs, many of which are

served by or built over similar technological infrastructures governed by technology giants such as Amazon, Google, Microsoft, or Cisco.

In the face of intensifying forces that stimulate convergence of actors around similar practices and preferences, we particularly probe the following questions:

1. Is it possible to design a generic business model framework for PBTA in order to capture and respond to the pressures of convergence?
2. How do PBTA around the globe perceive this ever-increasing and arguably homogenizing pressure on their current business models? What are the basic themes of convergence?
3. Which local/contextual factors cause divergence of PBTA' responses to such challenges?
4. What should PBTA do to put the designed model in practice in order to respond to the challenges posed by convergence/divergence?

Motivated by the former questions, our study aims initially to design a generic business model for PBTA experts coupled with external contextual determinants and then explore convergence/divergence of their responses based on a fuzzy logic methodology. Furthermore, the study also includes a deployment methodology for PBTA to respond to the external challenges by scalable solutions through agile steps. By doing so, our study contributes to the design and deployment of a generic business model framework specifically geared to the interests of PBTA and illustrates how attributions of importance to the external and internal components of the model vary across different metropolitan areas around the globe. The study extends business model literature by introducing specific mechanisms to capture convergent and divergent contextual factors and create immediate and long-run response sets to such challenges. It also contributes to the theoretical discussions of convergence and divergence by illustrating the degree of convergence perceptions on the particular material and social factors in the transportation domain.

Business models represent important vehicles in the substantiation of strategies [13] and offer public, private, and non-governmental enterprises significant advantages in designing, delivering, and monetizing their value propositions [14]. Driven by iterative testing and agile response to rapid changes occurring in technologically complex and competitive environments, the use of Business Model Canvas (BMC) has become prevalent in large-scale enterprises and non-governmental organizations alike [15,16]. Although the use of BMC framework has been previously shown in the transportation domain such as for shared mobility services [17–22], and Mobility-as-a-Service platforms [23,24], there has not been any attempt to design a generic BMC for a PBTA in the literature. The design and adoption of this model have become necessary as increasing numbers of technology and service platform providers, all of which run and scale their operations on a BMC framework, will be more likely to integrate and align their operations with a PBTA. Besides, rapid transformations in the consumption patterns of metropolitan citizens in the form of faster, safer, cleaner, and better-quality mobility will be more likely to pressure PBTA to align their value propositions with these demands in a more agile way [25]. Thus, there will be additional benefits for the top management teams of PBTA to adopt BMC as a tool to efficiently integrate with high-technology service providers as well as meet the escalating demands of their passengers. Moreover, collective adoption and use of BMC by PBTA will enable the development of cross-border comparative analyses and effective sharing of better fitting solutions among the PBTA of global metropolitan areas.

The study adopts the GUEST framework offered by Perboli and his colleagues [26], which includes five consecutive steps: go, uniform, evaluate, solve, and test. Based on the framework, a generic model is defined at the first step, which incorporates internal and external layers. The internal component of the model is composed of a conventional nine-block BMC enhanced with an additional block accounting for the impact. The external layer of the model, on the other hand, includes a conventional situation analysis tool: the PESTEL (Political, Economic, Social, Technological, Environmental, and Legal) framework is customized for PBTA. A comprehensive literature survey and the professional expertise of authors are used in the development of the generic model. At the second step, a

consultation with PBTA experts around the globe is made in order to assess the usability of the framework in terms of the degree to which each component contributes to the performance of an ideal PBTA operation. Since different attributions of importance for various model components have been made by experts from different countries, it is decided that the focus of the analysis should be on global convergence and divergence of different components. At the third step, the collection of data from different metropolitan cities across the globe is organized with the Delphi method, which is used as an input for a fuzzy-logic analytic hierarchy process (fAHP) to explore the importance of model blocks. At the fourth step, we collected linguistic data, translated the data into fuzzy measures, and ran the analysis to identify the convergence/divergence of importance attributions among different global experts. To complete the validation of the model, a follow-up consultation with global PBTA experts is conducted. Following the consultation, two actionable deployment templates are created, one for immediate response to an abrupt external pressure and the other for a long-run systematic response to ever-changing external pressures. By designing an actionable generic BMC framework for PBTA, the study not only becomes the first to offer a cross-national strategic analysis and action tool for PBTA in the face of increasing global pressures but also fills a significant gap in the BMC literature by presenting how strategic responses to changes in one or more components of the external environment should be made within a BMC.

The paper is structured in four main sections. The following part of the first section introduces the theoretical assessment of the convergence–divergence debate and discusses the development of the BMC idea. In the second section, model development, method selection, application, data collection, and data analytic procedures are discussed. The third section discusses the findings in terms of their implications for the acceptance of the proposed model by PBTA, convergence/divergence debate, and practical guidelines for PBTA. The study concludes by underlining its contributions to the convergence/divergence debate and potential implementation of the proposed model in the public transportation domain.

*1.2. Theoretical Background of Convergence and Divergence Theses*

The convergence–divergence debate has been traversing diverse theoretical domains like sociology [27], politics [28], economics [29], methodology [30], and technology [31–35] for a long time. Globalization processes lie at the center of the debate. Proponents of convergence argue that globalization, especially after the collapse of the Soviet Regime, stimulated increasing homogenization around a democratic and liberal market regime [29], which in turn, accelerated similar consumption patterns and technological processes/products across countries [36]. Scholars, who contest these ideas, assert that historical processes shaping geographically diverse settings such as culture, institutions, resource endowments, and economic development patterns inhibit convergence around a single governance system and practices [27,37]. Thus, proponents of the divergence thesis argue that nation-states have developed dissimilar systems of economic organization and such systems only converge to the extent that they share underlining social, economic, and political institutions [27,38]. Even though scholars provided significant case-based evidence against convergent political and economic behavior across regions and nations [38], recent studies provide repeated support in favor of technological convergence as digital processes have increasingly been built upon the Internet (single-net) architecture. Furthermore, with the advances in mobile ICT, instantiation of access to and concurrent-collective use of applications across nations became possible and widespread across the globe [36]. Accelerated convergence of service technologies across the globe significantly homogenized values, lifestyles, and consumption patterns for many, who became connected to each other on popular platforms. Yet, the limits to convergence in the form of local regulations, political regimes, values, and institutions still persist [39].

Public transportation is not an exception in this debate, as scholars found similar use of practices and technological artifacts/services across different metropolitan cities [40].

Shared, connected, and intelligent mobility systems became prevalent across many cities around the world often under the same labels [32,41,42]. Thus, PBTA are faced with dual pressures stemming from increased technological convergence and divergent forces of local transportation regulations, conventions, and economic affordances. However, there is no systematic scholarly effort that investigates whether forces stemming from globalization stimulate convergent public transportation practices across different regions or if there are specific limits to such convergence.

*1.3. Theoretical Assessment of Business Model Canvas*

Taking convergent and divergent pressures into account, especially considering the intensity and pace of technological transformation, performing conventional strategic, and political analyses (i.e., SWOT (Strengths, Weaknesses, Opportunities, and Threats) and PESTEL methodologies) will be less useful for PBTA. These analyses require comprehensive plans to be prepared for long periods, which in turn need a number of future scenarios to be prepared for possible contingencies. Even when such analyses are made, they provide little room for solidifying unique value propositions for targeted stakeholders within an integrative perspective and do not allow decision-makers to account for possible interactions between complementarities, substitutes, access opportunities, and revenue streams. Such comprehensive and complex plans almost always fail under turbulent conditions. Failure, on the one hand, consumes time and valuable resources, and on the other hand, it demotivates stakeholders.

Agile and iterative steps to put forward hypotheses about key business model assumptions, designing fast validation actions to test these hypotheses, and iterative construction of this cycle towards a sustainable and repeatable business model is the new way offered by lean business model development [43]. Built on the former logic, Blank and his colleagues [44] popularized the use of BMC in high-technology startups, as they used BMC as a baseline instrument to accompany and diffuse the philosophy of agile development across all functions and processes of a prototypical enterprise. Accordingly, the use of BMC as a tool to shape strategic action has increased rapidly in the last few years. Although the coinage of the business model concept dates back to 1957 [45], its adoption and use significantly increased in technology-intensive firms after the introduction of BMC by Osterwalder [46].

BMC represents a significant breakthrough in the systematic modeling of value configuration, delivery, and commercialization because it is technology agnostic, user-friendly, comprehensive, and focuses on the interaction between components of the model [47]. The BMC allows mapping the interactions between resources and competencies of an enterprise with potential customer segments, channels, and value propositions as well as their financial outcomes. The comprehensive and interactive framework of the BMC offers significant advantages in terms of designing, testing, and transforming the way enterprises operate their businesses.

Originally the BMC includes nine blocks, under three major pillars. The first pillar is about value creation and it includes (i) the detailed definition of targeted customer segments, (ii) the channels by which these segments will be served, (iii) specific ways by which relationships with potential and earned customers will be developed, nurtured, and maintained, and (iv) value propositions offered to each customer segment. The second pillar is composed of activities designed for delivering the value created in the first pillar by (v) detailing which key resources will be used, (vi) which key activities will be performed and (vii) which of these activities will be outsourced to partners. The third pillar is about the monetization of the created value and its costs, which includes (viii) the revenue model and (ix) the cost structure of the resources and activities used, including the effects of outsourced partnerships. The integrative structure of the BMC as well as its iterative and agile use in technology-based companies [48,49] enables its adoption in PBTA as viable strategy development and implementation tool. Even though BMC has been previously applied in the transportation domain, specifically in shared mobility services [17–19,50–53],

Mobility-as-a-Service platforms [22–24], and green transportation [54–57], it has never been applied for a PBTA. With the intensifying globalization of consumer preferences and the diffusion of ICT platforms across borders, the pressure on the management teams of PBTA about the faster and more accurate responses to these challenges will increase, compelling them to adopt agile and comprehensive strategic action tools like BMC. In the next section, a generic BMC for a PBTA is designed and customized according to its operations together with external environmental forces to account for intensifying global demands.

## 2. Methodological Framework

### 2.1. Model Design

At this point, there is a need to revisit the original BMC for its adoption by a PBTA, since PBTA perform public service and do not solely operate on the logic of profit maximization. As BMC has also been designed for non-governmental organizations or social enterprises in the literature [58–61], we have thoroughly analyzed alternative BMCs in terms of their ease of application and the degree to which they balance revenue and impact perspectives. Sustainable business models need to address both internal elements such as the efficient and responsible deployment of resources and activities as well as external stakeholder relations. It is impossible to sustain an organization without taking its external stakeholder relationships into account. Therefore, initially, a literature search was conducted to analyze how scholars have until now incorporated the sustainability dimension in the BMC. For example, Daou et al. [62] introduced Ecocanvas Business Model for a circular economy, emphasizing environmental and social foresight dimensions. Whereas the environmental foresight in their model is built on the PESTEL tool, which represents Political, Economic, Social, Technological, Environmental, and Legal factors, the social component deals with the transformation of social structure, networks, and habits [62]. Other scholars added another layer of social-environmental costs and benefits under the monetization pillar of the traditional BMC to emphasize public goals [58]. Sanderse also added an impact block under the monetization pillar to adjust the use of BMC for non-for-profit enterprises [60]. Following these studies, a generic BMC customized for the operations of a PBTA is designed and illustrated in Figure 1. In order to balance the economic costs and benefits of a PBTA with social and environmental costs and benefits, an additional component of impact is added into the monetization pillar of the traditional BMC.

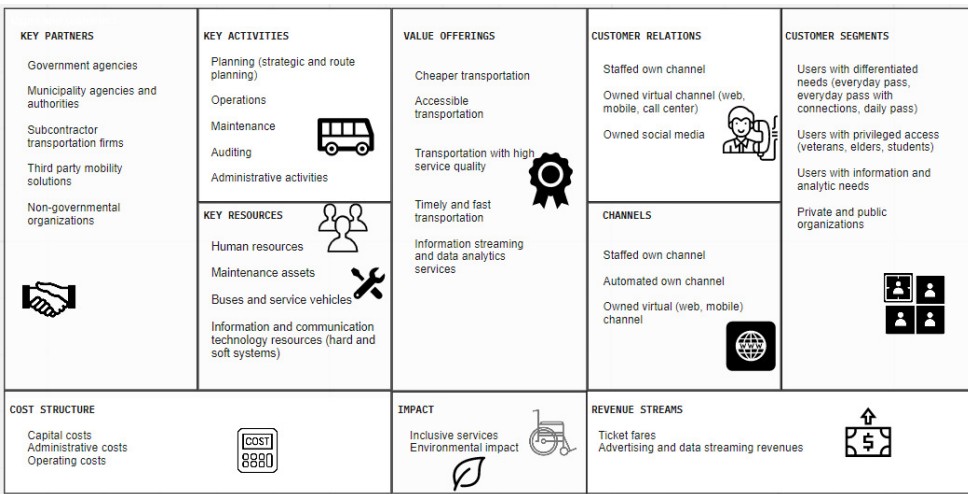

**Figure 1.** Business model framework for public transportation authority regarding impact analysis.

The first pillar of the generic model incorporates customer segments, value offers, channels, and customer relation blocks. The customers of a PBTA bus operation are often segmented according to dissimilar use patterns (i.e., faster or longer routes, or tourists, who use transportation for a limited time) or privileged access granted to certain societal groups (i.e., veterans, teachers, students, elderly, women with children, etc.).

Thus, value propositions should be geared to these segments in terms of providing timely, cheap, accessible, and flexible transportation routes. As information processing becomes indispensable with the recent progress of ICT, inboard and outboard vehicle information processing and broadcasting offerings complement these traditional value propositions. Such information processing and visualizing technologies often take the form of timetable, route, and streaming services accompanied by inboard or outboard entertainment and advertising offers. Many PBTA still operate physical channels in the form of location-based ticket offices, either with manned or automated technologies. Physical channels, especially staffed ones, still constitute an important traveler relationship management resource. However, with the advanced ICT solutions, many PBTA complement their physical staffed services with mobile applications, owned social media, automated or manual electronic mails, chat-bots, and toll-free call center solutions [13].

The second pillar of the generic model includes key resources, activities, and partnerships. As resources, PBTA employ large-scale physical assets in the form of vehicles, tooling, and machinery for vehicle maintenance and road assistance, a variety of hard and soft systems to operate, monitor, and support complex traveling services, and crowded staff to govern all these activities at various levels. Accordingly, key activities of a PBTA can be defined as planning, operating, maintaining, monitoring, and governing all fleet operations as well as support activities to grow and keep the efficiency basis of all operations at optimum levels. With the advancement of ICT, data collection, storage, transmission, processing, and visualization activities have increased their shares in the traditional operational activities of a PBTA [14]. PBTA traditionally collaborate with key partners such as municipalities and other authorities to govern all travel activities within a metropolitan area. Besides, many PBTA outsource their fleets to key private parties for efficiency and cost reduction purposes [25]. Recently, some PBTA have started to integrate with a variety of private or public mobility services (bikes, scooters, taxis, or shared-ride platforms) [63].

Consequently, the third pillar of the model is about a sustainable revenue stream that covers revenues, costs, and impact blocks. In terms of revenues, it is possible to argue that PBTA are still strongly dependent on ticket fares for their services [64]. The share of additional revenues streaming from advertising, data, and service partnerships remains relatively small. Costs depend on operational expenses such as labor, fuel, maintenance, and fixed costs of assets (vehicles and maintenance operations). Although electrification seems to challenge some of the costs, it is still too early to transform the cost base of PBTA for the near future. Since PBTA do not solely operate for profit, impact objectives should also be considered here. With the environmental sustainability pressures increasing, PBTA will have to consider a more sustainable operation and try reducing their carbon footprint [65]. Second, they have to be inclusive in their operations, meaning that they need to meet the transportation needs of all citizens (extended routes, flexible scheduling, etc.) and provide positive discrimination to needy citizens (elders, veterans, women with small children, students, low-income segments, etc.) [66].

As the BMC includes only primary value chain activities and partners such as suppliers, channels, key customers, and complimentary service/good providers within its scope, it is necessary to integrate other external stakeholders, as their pressures for legitimacy and performance are critical in the long-run survival. Scholars also advise complementing the BMC with external analysis using a PESTEL approach [67]. To this end, a systematic literature survey is conducted to construct a generic external analysis of a PBTA.

To address political criteria, regulation, tax policy, labour law, and government structure are taken into account for PBTA operations, as these factors are found to shape many of the business model blocks for public transportation [68]. To account for economic factors, petrol price, consumer disposable income, private car ownership, gross domestic product, average wage, exchange rate, inflation rate, transportation budget, and currency stability are considered, as they have been frequently addressed in the literature to influence public transportation operations [68–70]. For social factors, growing population, aging population, mobility, and transportation culture, and lifestyle are considered as key elements to shape

public transportation [68,70]. For addressing the impact of technological factors, rapid development in technological fields and faster innovation cycles are taken into account [68]. For explaining the legal factors, scholars emphasized the importance of government regulations, health and safety law, and data protection. Consequently, for environmental factors, environmental protection and climate change imperatives are most commonly discussed in the transportation domain [68]. Even though the influence of external factors mainly grouped under the rubric of PESTEL is extensively studied in the transportation domain, global differences or similarities of these factors across countries and links between the PESTEL framework and BMC have been overlooked. To address these gaps, the following parts introduce the data collection process, followed by the analysis method and key findings.

*2.2. Data Collection*

The generic BMC and its external environment for a PBTA have been proposed in the previous section based on the literature and expert opinions. To understand if the proposed generic model fits the current operational needs of PBTA and represents successful and efficient operational credentials, members of IBBG (International Bus Benchmarking Group) experts are consulted. The experts were selected with respect to their experience in the public transportation sector, which required a minimum of ten years of experience. Besides, for the experts who were working as managers, being employed in different departments such as planning, operation, innovation, strategy development, and finance were preferred to create a holistic perspective in the evaluation of the developed business model. In summary, the survey design is framed regarding MCDM methods [71].

The study was conducted by national and international transport experts around the world. To fill the comparison matrices, empty comparison tables were sent via e-mail with a textual explanation of the proposed model, and online meetings were held between the corresponding author and IBBG members. Since explanations of the factors are complex, additional support was also provided via the exchange of e-mails and online visual communication to explain each factor and explicitly clarify how the designated model worked. The data were collected from sixteen experts, who represented different metropolitan PBTA. Cities were chosen from four continents of the world, which were selected from the list of megacities according to the definition of the United Nations [72]. To represent megacities across the globe, a minimum population of 500,000 inhabitants or more was set as another selection criterion. The distribution of experts, metropolitan cities, their PBTA organization types, and their representative continents are displayed in Table 1. The great majority of cities have both authority and operator functions such as Dublin, Istanbul, Kuala Lumpur, Montreal, Moscow, and New York.

**Table 1.** Bus organization types of cities.

| Continent | City | Authority | Operator |
|---|---|---|---|
| Europe | Barcelona | - | x |
| | Dublin | x | x |
| | Istanbul | x | x |
| | London | x | - |
| | Moscow | x | x |
| America | Montreal | x | x |
| | New York | x | x |
| | Vancouver | - | x |
| Asia | Kuala Lumpur | x | x |
| | Singapore | - | x |
| Australia | Sydney | - | x |

During the initial consultation with IBBG members, we are alerted by potential variation across different international members in terms of different emphasis of model compo-

nents over others. Accordingly, a formal analytic method is sought in order to compare and contrast varying importance levels attributed to the components of the model.

*2.3. Method*

To assess different attributions of importance to the model components at the sub-criteria level, a systematic analysis of multi-criteria-decision-making (MCDM) methods in the literature was conducted. Among many different MCDM methods in the literature, which include Analytic Hierarchy Process (AHP), Analytic Network Process (ANP), Case-based Reasoning, Data Envelopment Analysis, Simple Multi-Attribute Rating Technique, Goal Programming, Elimination and Choice Translating Reality (ELECTRE), Preference Ranking Organization Method for Enrichment Evaluation (PROMETHEE), Simple Additive Weighting, and Technique for Order of Preference by Similarity to Ideal Solution (TOPSIS) [73], AHP is ranked at the top in terms of its use and adoption in different domains [74].

AHP operates on pairwise comparisons with respect to judgments of experts to get priority scales [75]. In the classical AHP method, experts' evaluations are based on crispy numbers. Due to the complexity of human thinking, quantitative judgments, and numerical data can be inadequate in many circumstances. Thinking about abstract concepts like BMC and PESTEL, it is very hard if not impossible for an expert to make numerical comparisons between different components of a comprehensive model. Thus, AHP scholars recently incorporated linguistic expressions of attribution into crispy comparisons within a fuzzy set environment [76]. Using fuzzy sets, one can model characteristics of obscurity and approximate under uncertain conditions. In transportation studies, fuzzy sets were applied in different domains including optimization [77], sentiment analysis [78], traffic flow modeling [79] and, safe transportation [80]. Also, it was conducted for business efficiency [81], e-learning [82], seismic vulnerability assessment [83], information technology governance evaluation [84], and supplier evaluation [85]. Accordingly, the AHP method has been extended to several fuzzy sets which are summarized in the study by [86].

The fuzzy environment which was introduced by [87] is used in determining the importance of the main and sub-criteria of the developed business model framework. The Spherical Fuzzy AHP (SF-AHP) method is employed in the study due to assigning the parameters of that membership function with a larger domain [88]. Using the hesitancy degree, we deal with the vagueness of the problem in decision-making [88,89]. In addition, in real life, the sum of the degree of memberships can exceed 1. For these situations, Spherical Fuzzy Set (SFS) can be used as an extension of Picture Fuzzy Set (PFS) [90]. SFS preliminaries were conducted in this study which was introduced by [86].

The application steps of the proposed model are represented in detail as below.

Step 1: Determine the main and sub-criteria of the public transportation business model with respect to the hierarchical structure.

Step 2: Constitute pairwise comparisons using Spherical Fuzzy Sets judgment matrices based on the linguistic terms which are represented in the study by [86] that are shown in Table 2.

**Table 2.** Spherical fuzzy sets linguistic terms [86].

| Priority in Pairwise Comparisons | $\mu, \nu, \pi$ | Score Index (SI) |
|---|---|---|
| Absolutely more importance | (0.9, 0.1, 0.0) | 9 |
| Very high importance | (0.8, 0.2, 0.1) | 7 |
| High importance | (0.7, 0.3, 0.2) | 5 |
| Slightly more importance | (0.6, 0.4, 0.3) | 3 |
| Equally importance | (0.5, 0.4, 0.4) | 1 |
| Slightly low importance | (0.4, 0.6, 0.3) | 1/3 |
| Low importance | (0.3, 0.7, 0.2) | 1/5 |
| Very low importance | (0.2, 0.8, 0.1) | 1/7 |
| Absolutely low importance | (0.1, 0.9, 0.0) | 1/9 |

Step 3: To calculate the consistency ratio (CR), convert the linguistic terms to corresponding score indices in the pairwise matrix. Then apply the classical consistency check for each pairwise comparison matrix that is presented in Table 3 [75]. If CR is less than 10 percent, pass Step 4, otherwise, go back to Step 2 and re-evaluate the matrices.

**Table 3.** Random consistency index (RI) [75].

| Size of Matrix | 1 | 2 | 3 | 4 | 5 | 6 | 7 | 8 | 9 | 10 |
|---|---|---|---|---|---|---|---|---|---|---|
| RI | 0.00 | 0.00 | 0.58 | 0.90 | 1.12 | 1.24 | 1.32 | 1.41 | 1.45 | 1.49 |

Step 4: Determine the Spherical Fuzzy Sets weights of criteria using the Spherical Weighted Arithmetic Mean (SWAM) operator which is provided in the study by [88].

Step 5: Defuzzify the obtained weights of each criterion using the score functions related to Spherical Fuzzy Sets that are given in the study by [88].

Step 6: Find global weights for each level according to the hierarchical layer.

Step 7: Rank the criteria concerning the defuzzified global scores. The largest global score means the most important criteria in others.

*2.4. Findings*

As one of the aims of this study is to answer if PBTA experts around the world ascribe similar or divergent weights to the components of the model designed in the previous section, analyses were run in the data set gathered from sixteen PBTA experts based on a Spherical Fuzzy AHP (SF-AHP) procedure. The hierarchical structure of the model is illustrated in Figure 2.

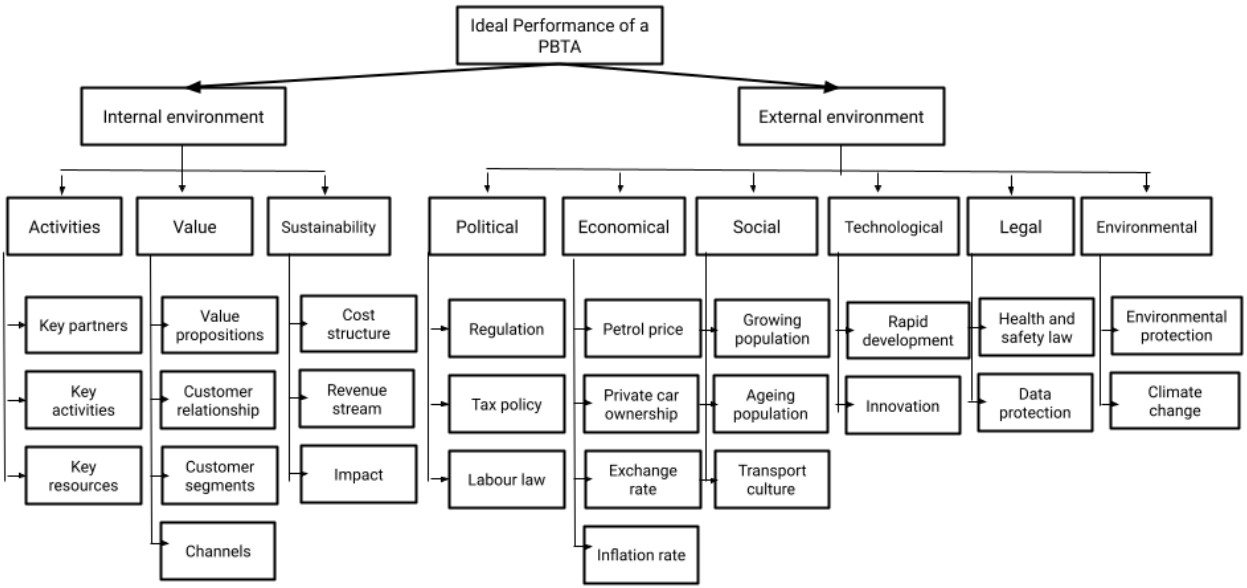

**Figure 2.** The proposed model for a successful PBTA operation.

Each expert's opinion is considered equal in the study. Due to space constraints, all of the calculations, which were made to attain spherical and crisp weights for all levels and experts, were not included in the main body of the study. However, we did include an example, containing the results of experts 1, 2, and 16 for the first level, displayed in Table 4. All of the calculations are available on request from the corresponding author. To understand clearly, abbreviations of the main and sub-criteria are given as follows. IE: Internal Environment, EE: External Environment, AC: Activities, VA: Value, SU: Sustainability, PO: Political, EC: Economical, SO: Social, TE: Technological, LE: Legal, EN: Environmental, KP: Key partners, KA: Key activities, KR: Key resources, VP: Value propositions, CR: Customer rela-

tionship, CS: Customer segments, CH: Channels, CS: Cost structure, RS: Revenue streams, IM: Impact, RE: Regulation, TP: Tax policy, LL: Labour law, PP: Petrol price, PC: Private car ownership, ER: Exchange rate, IR: Inflation rate, GP: Growing population, AP: Ageing population, TC: Transport culture, RD: Rapid development, IN: Innovation, HL: Health and safety law, DP: Data protection, EP: Environmental protection, CC: Climate change.

**Table 4.** Pairwise comparison of the main criteria.

| Experts | Main Criteria | IE | EE | CR | SWAM | | | Local Weights |
|---|---|---|---|---|---|---|---|---|
| Expert 1 | IE | EI | SMI | 0.554 | 0.400 | 0.351 | | 0.557 |
| | EE | SLI | EI | 0.454 | 0.490 | 0.358 | | 0.443 |
| Expert 2 | IE | EI | SLI | 0.454 | 0.490 | 0.358 | | 0.443 |
| | EE | SMI | EI | 0.554 | 0.400 | 0.351 | | 0.557 |
| Expert 16 | IE | EI | HI | 0.618 | 0.346 | 0.303 | | 0.610 |
| | EE | LI | EI | 0.417 | 0.529 | 0.331 | | 0.390 |

Finally, Spherical fuzzy global weights of the model's main and sub-criteria according to each continent with their means and standard deviations are presented in Table 5. Findings indicate several important points in terms of understanding the degree to which experts converge and diverge in their attributions of importance to the components of our generic model. First, as it is disclosed in Table 5, for all continents business model components outweigh external environmental components at the first level. Second, excluding Asian metropolitan cities, all global experts also converged in their importance attributions about the second level components of the business model. Experts from European, American, and Australian metropolitan cities rated key resources and activities over sustainable revenue streams and value offerings respectively, whereas Asian respondents rated sustainability priorities and value offerings as more important than key resources and activities. Divergence stands out more when one considers the attributions of importance to the components of the external environment. Here, Australian experts' considerations about economical, legal, technological, and environmental factors diverge from other experts' opinions from different continents. According to these results, political concerns represent key priorities for European and Australian experts, economic concerns are generally found critical for Europeans, innovation intensity is highly taken into account by Asian experts, and legal issues seem to concern American experts more than others. While there is relatively less dispersion on the social environmental factors across continents, there is higher dispersion about the importance attributed to the economical and technological factors.

**Table 5.** Global weights regarding continents from the fuzzy approach with mean and standard deviation.

| Level | Criteria | Europe | America | Asia | Australia | Mean | Standard Deviation |
|---|---|---|---|---|---|---|---|
| First level | IE | 0.582 | 0.607 | 0.613 | 0.659 | 0.615 | 0.032 |
| | EE | 0.418 | 0.393 | 0.387 | 0.341 | 0.384 | 0.032 |
| Second level | AC | 0.218 | 0.259 | 0.182 | 0.289 | 0.237 | 0.047 |
| | VA | 0.179 | 0.168 | 0.202 | 0.185 | 0.183 | 0.014 |
| | SU | 0.185 | 0.180 | 0.229 | 0.185 | 0.195 | 0.023 |
| | PO | 0.079 | 0.073 | 0.058 | 0.079 | 0.072 | 0.010 |
| | EC | 0.074 | 0.051 | 0.066 | 0.039 | 0.058 | 0.016 |
| | SO | 0.064 | 0.057 | 0.057 | 0.062 | 0.060 | 0.004 |
| | TE | 0.074 | 0.074 | 0.076 | 0.059 | 0.070 | 0.008 |
| | LE | 0.066 | 0.073 | 0.068 | 0.051 | 0.065 | 0.009 |
| | EN | 0.061 | 0.065 | 0.062 | 0.051 | 0.060 | 0.006 |

**Table 5.** *Cont.*

| Level | Criteria | Europe | America | Asia | Australia | Mean | Standard Deviation |
|---|---|---|---|---|---|---|---|
| | KP | 0.082 | 0.101 | 0.063 | 0.112 | 0.090 | 0.022 |
| | KA | 0.062 | 0.079 | 0.056 | 0.088 | 0.071 | 0.015 |
| | KR | 0.074 | 0.079 | 0.063 | 0.088 | 0.076 | 0.011 |
| | VP | 0.053 | 0.049 | 0.055 | 0.051 | 0.052 | 0.003 |
| | CR | 0.045 | 0.046 | 0.050 | 0.032 | 0.043 | 0.008 |
| | CS | 0.043 | 0.027 | 0.053 | 0.049 | 0.043 | 0.011 |
| | CH | 0.038 | 0.045 | 0.043 | 0.053 | 0.045 | 0.007 |
| | CT | 0.074 | 0.067 | 0.084 | 0.070 | 0.074 | 0.008 |
| | RS | 0.064 | 0.058 | 0.060 | 0.045 | 0.057 | 0.009 |
| | IM | 0.047 | 0.055 | 0.084 | 0.070 | 0.064 | 0.017 |
| | RE | 0.027 | 0.031 | 0.019 | 0.023 | 0.025 | 0.005 |
| | TA | 0.024 | 0.024 | 0.021 | 0.028 | 0.024 | 0.003 |
| | LL | 0.028 | 0.018 | 0.019 | 0.028 | 0.023 | 0.006 |
| Third level | PP | 0.021 | 0.012 | 0.017 | 0.008 | 0.014 | 0.006 |
| | PC | 0.018 | 0.007 | 0.014 | 0.010 | 0.012 | 0.005 |
| | ER | 0.018 | 0.015 | 0.019 | 0.010 | 0.016 | 0.004 |
| | IR | 0.017 | 0.017 | 0.016 | 0.011 | 0.015 | 0.003 |
| | GP | 0.023 | 0.027 | 0.023 | 0.020 | 0.023 | 0.003 |
| | AP | 0.019 | 0.017 | 0.013 | 0.015 | 0.016 | 0.003 |
| | TC | 0.021 | 0.014 | 0.021 | 0.027 | 0.021 | 0.005 |
| | RD | 0.043 | 0.046 | 0.031 | 0.039 | 0.040 | 0.006 |
| | IN | 0.031 | 0.028 | 0.045 | 0.020 | 0.031 | 0.010 |
| | HS | 0.036 | 0.036 | 0.034 | 0.020 | 0.031 | 0.008 |
| | DP | 0.031 | 0.036 | 0.034 | 0.031 | 0.033 | 0.003 |
| | EP | 0.035 | 0.038 | 0.035 | 0.023 | 0.033 | 0.007 |
| | CC | 0.026 | 0.027 | 0.027 | 0.028 | 0.027 | 0.001 |

To validate the results of the study, a final round of consultation has been made with a limited number of IBBG experts. Whereas experts confirmed the viability and novelty of the model and expressed positive opinions about the deployment of the model as a strategy tool, they also underlined the fact that a more precise methodology is needed to guide PBTA about how to deploy BMC and take corrective action in the face of convergent/divergent environmental pressures. Considering the lack of knowledge repositories about the BMC framework and its deployment in the PBTA domain, a stepwise methodology to guide strategic action, which is associated with relevant environmental factors for varying temporal constraints, is also designed. The steps of the designed strategic action methodology are provided in the discussion part.

## 3. Discussion

### 3.1. Viability and Sustainability of the Proposed BMC for PBTA

An increasing number of studies have been published in different domains of literature about the sustainability of business models, which can be broadly categorized under reviews, conceptualization, and application [91–93,93–109].

Sustaining a business model is as key of a point as constructing it. Based on the aforementioned studies on the sustainability of business models, sustainability can be conceptualized both for external and internal elements of a business model. Whereas the internal elements are mostly related to the survival of organizations in terms of continuous and efficient supply of necessary resources: i.e., human, data, physical, financial, the external elements include maintaining external stakeholder relations: i.e., customers, community, government, partners to be perceived as legitimate and effective alongside with preserving non-human elements of nature. As the BMC originally proposed by [46] incorporates external elements of an enterprise only through customer segments and key partners, a more systematic treatment and integration of external elements into the BMC is vital for addressing sustainability pressures. Accordingly, the design of our model not

only includes an impact component to extend the conventional BMC for the integration of social inclusion and environmental impact [102,110–112] but also the inclusion of PESTEL into our framework systematically combines the missing external stakeholders and their demands for legitimacy and performance [58–61].

Many PBTA have already begun offering special services in order to increase their social impact. For example, IETT, which is the PBTA in Istanbul, introduces special bus routes for trekking, organic bazaars, and hospital lines regarding COVID-19 [113,114]. MTA, which is the PBTA of New York, and Coast Mountain, which is the bus company of Vancouver, offer rapid transport services for disabled and elderly people [115,116], whereas Transport for London (Tfl), which is the transportation authority in London, has been operating school routes [117]. Concerning the environmental impact, Tfl has just started to include hydrogen buses into the fleet for a pilot project [118]. Besides, a massive electrification plan for fleets is already underway for different PBTA such as Tfl, RATP (PBTA for France), Dublin Bus (PBTA for Dublin), Moscow for Transport (PBTA for Moscow), and SMRT (operator for Singapore) [119–123].

While repeated service offers by PBTA, which have positive social and environmental impact potential, are important, it is more critical to sustain such offers in order to ensure their continuity, efficiency, and greatest impact for most stakeholders. Adoption of a BMC framework integrated with external components, in this sense, might improve such service propositions by offering the possibility of an extensive search for demand, iterative and agile service design cycle, and sustainable operational capability, each integrating with and complemented by action steps. Based on qualitative consultations with global PBTA experts, it is encouraging to witness that they also understand and support the framework offered in our model. Furthermore, findings gathered from the formal evaluation of the proposed model indicate that the average level of impact (mean = 0.064) ascribed to the impact component is ranked second after cost structures (mean = 0.074), receiving more attention than revenue streams (mean = 0.057). Furthermore, the overall importance weight for the external components (mean = 0.384) is also significant across continents. Thus, it can be argued that the inclusion of an impact component within the BMC and integrating it with external components PBTA in order to emphasize external stakeholder pressures has been predominantly accepted, as it has received significant weight among global PBTA experts.

When the overall design and validation of the model is a significant contribution, the diffusion and adoption of it by PBTA across the globe remains a challenge. In order to ease absorption of the new framework, the design of strategic action steps for possible adoption of the designed model is discussed in the subsection which includes strategic action steps for PBTA.

### 3.2. Convergence and Divergence of the Model Components

The results gathered from the global experts about the pillars of the proposed BMC point towards important implications. First, the findings indicate that the internal components of the model have received much higher importance levels than the external environmental factors. Such consolidation of attributions is in line with the resource-based view of the firm, which affirms that different configurations of firm resources and capabilities are more important in determining firms' performance variations compared with external environmental factors, which include macro factors, competition, and substitutes [124]. Therefore, experts imply that they are not passive actors, who just try adapting to environmental changes but may proactively reconfigure their resource bases either to align more efficiently with future environmental variations or even shape them.

Moreover, experts from different continents agree more on the importance of the value propositions, cost structures, key resources, and channels components of the BMC. The discrepancy of the weights attributed to the former components is relatively low. On the contrary, the experts' scores tend to diverge for the elaboration of the importance of customer segments, key partners, and impact. First, Asian experts ascribe relatively lower

levels of importance to partnerships compared with American and Australian experts. On the other hand, Asian experts again seem to diverge from their counterparts in terms of attributing more weight to the cost efficiency and impact components of the BMC. Regarding the BMC, another important divergence is found in the degree of importance American and Asian experts ascribe to the customer segments. Asian authorities seem to place significantly more importance on customer insight than their American counterparts. However, more research is needed to account for why such a difference exists among the perceptions of experts from different continents. As the proposed model required experts to give comparative ratings, existing differences between the governance structures, resources, and capacity levels and repository of partnerships of participating PBTA are likely to stimulate divergence among perceptions. Besides, the lingering effects on COVID-19 both on the European and American continents should be taken into account as well. Since the negative impact of the pandemic is much higher on the metropolitan cities located in the latter continents, it seems logical to assume that PBTA in these continents shift their focus more on the cost-efficiency of their operations rather than differentiating and aligning their value offerings.

Relatively more divergence exists in experts' attributions of importance to different external environmental factors. Considering that the nation-states still shape legal, economic, and political frameworks as sovereign organizations, variations among these dimensions across different continents seem plausible. Besides, the both national systems approach [27] and varieties of capitalism approach [125] introduce convincing arguments about the institutional barriers that fuel the divergence of practices and societal structures across the globe. Scholars who argue for the divergence thesis generally underscore the lingering differences in values and norms, which are shaped by macro-level institutions governing the society [37]. Contrary to such claims, experts from different continents converge on their attributions of importance to social and environmental factors. This is interesting because the latter theories assert that trust and authority relations prevalent in societies and the propensity of integration among societal members represent important barriers to converge across nations [37]. Thus, convergence among experts' perceptions about societal factors signals a critical inclination toward a convergent road to similar norms, expectations, and lifestyles. Yet, significant divergence exists among many experts on their ascription of importance to economic factors, health and safety regulations, and innovation drivers. Observations about more divergence concerning the economic and technological aspects of the external components are contrary to the expectations of the convergence thesis as well since proponents argue that globalization pressures initially infiltrate into local preferences and behaviors through increased integration of markets and technological artifacts.

In this sense, the study contributes to the theoretical debates on the convergence and divergence by proposing that perceptions of external social components of a PBTA converge more than material components like economic and technological factors. Besides, abstraction levels play an important role in the level of convergence and divergence attributions as the degree of divergence becomes visible among the specific components categorized under the hierarchical layers of the model. Consequently, the perceptive nature of scores inevitably compelled respondent experts to consider their current state of resource and capability base, the extent of their partner networks, and customer services during the assessment of the potential impact of external factors. Here, affirmations of divergence thesis about the sociological nature of one's understanding and immersion of his/her action in the wider social and material environment are imperative.

### 3.3. Strategic Action Steps for PBTA

As the final consultancy with the IBBG experts underscored the necessity of a precise methodology, which explicitly defines how assessments of external components should be translated into strategic actions within the BMC framework, a stepwise methodology is designed based on the agile approach [43,126,127]. The steps for short-run and long-run

actions are distinguished as their action steps in the BMC framework diverge. According to the distinguished action steps, the former requires swift responses to abrupt and compelling changes occurring in the environmental components, whereas the latter requires continuous observation and strategic coupling. For short-run response conditions, an immediate regulation forcing PBTA to implement distancing measures such as halving passenger loads or prohibiting standing passengers on buses because of the COVID-19 pandemic can be given as an example. For long-run responses, faster digitalization trends such as growing penetration of mobile payment platforms or increasing private car ownership following the pandemic should be assessed as a long-term environmental change, which should be analyzed according to the second template. In order to assess, design, and implement action steps a task force should be created within the PBTA, which ideally is composed of diverse competencies, representing all of the value chain activities such as strategy, customer relations, operations, finance, R&D, and technology.

As illustrated in Figure 3, short-run response steps begin with a compelling external shock, which can be a sudden regulation change or a disaster situation. These external shocks require immediate attention and fast compliance. Therefore, task forces should first assess the impact of external shock on key activities and/or resource allocation arrangements. Even though gaining customer insight is not critical in immediate response conditions, an assessment of potential demand variations is necessary, as such changes can influence capacity and resource allocations. Since the objective of new designs is often clearer in these situations, as they are explicitly stated in regulations or public disclosures, converting current processes towards espoused processes takes priority. At the fourth step, the task force should assess whether to externalize the new processes to a third party or to implement it with internal resources based on feasibility and availability of partners. Complementing the former step, an analysis of revenue, cost, and impact estimations should be conducted. This analysis will guide future steps about the acquisition of new resources, changing resource allocation arrangements, or revising the process designs. At the same time, a strong communication program should be made to inform customers and other relevant stakeholders about the changes in value offers, channels, or payment systems, which might be influenced by the external shock. Finally, the task force should closely watch new process deployment and assess potential long-term effects of changes on customer preferences and habits. Once such analysis begins it triggers and becomes integrated with the long-run action template.

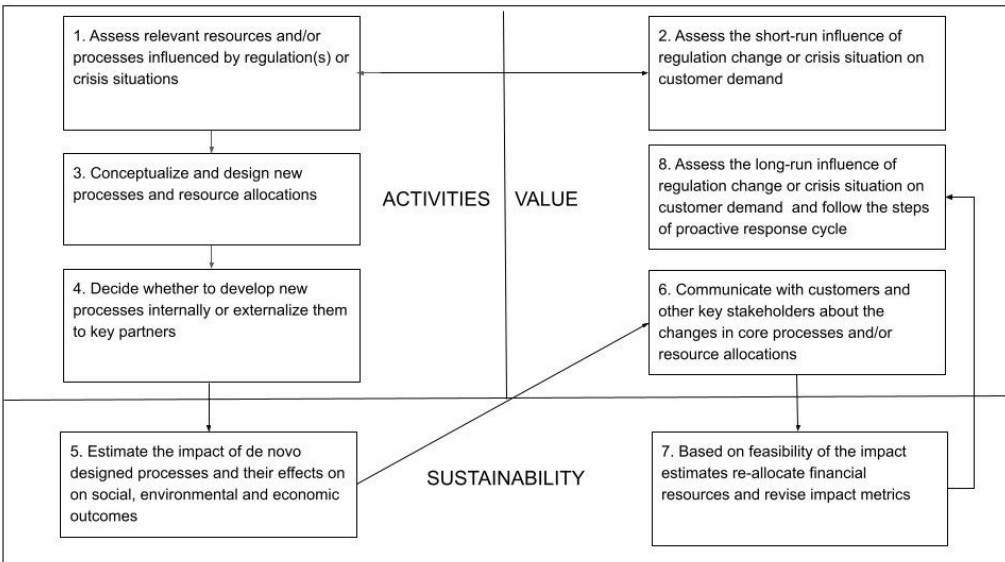

**Figure 3.** Short-run response action steps.

Long-run response steps, as illustrated in Figure 4, begin with the assessment of the long-term influence of converging/diverging external factors on the preferences and behavioral patterns of customers. The use of primary data from observations, event logs of customer relationship databases, and qualitative research like focus groups and interviews are advised in this step in order to gain maximum insight about changes. The second step is to design new services which would satisfy changing customer preferences. Since the value block of the BMC includes channels and customer relations along with customer segments and value offers, the design of services should incorporate customer acquisition, value delivery, and relational services as well. The next step is to decide whether the new service offer should be performed by internal resources and activities or it should be externalized to a third party. Once the decision of externalization is made, an assessment of the impact on costs, revenues, and social and environmental metrics should be made. If it is estimated that the net impact of new services is positive, service should be offered and the team should collect data from use patterns in order to complete continuous learning and development cycle. If feasibility analysis signals insufficient performance impact, the steps before the analysis should be reiterated in order to reach a feasible solution.

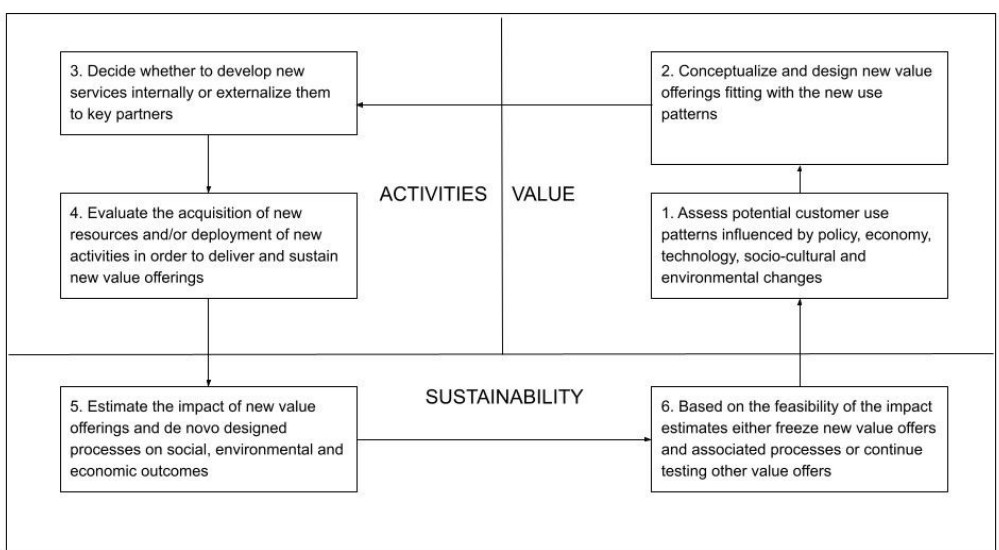

**Figure 4.** Long-run response action steps.

The proposed strategic action methodology which is differentiated for short and long-run responses is likely to increase the adoption and deployment of the proposed model. The model is flexible enough to be customized according to the demands of authorities and operators alike since value, activity, and sustainability pillars allow the integration of different operational and decision-making capacities. Besides, once the adoption of the model becomes easier its further diffusion and comparative implications will become available. The international level of convergence on the model will likely provide valuable data and opportunities to share best practices among the members of IBBG.

## 4. Conclusions

Building on the recent debate about the convergence of technological artifacts and politic-economic practices across the globe, the study aims to query the resonating impact of these homogenizing pressures on the public transportation domain by exploring the convergent and divergent responses of PBTA experts to a generic business model.

First, the designed model, which emphasized both internal and external sustainability dimensions, has received significant support from global PBTA experts. Both the inclusion of external factors and integration of an impact component to the original BMC by Osterwalder [46] have received significant importance weights as well as positive verbal responses from global experts. The results indicate that the designed model has an im-

portant potential to be used in the public bus transportation domain as a viable strategic action tool in the course of intensifying convergence of technological artifacts, markets, and consumer preferences. Thus, the study contributes to the public transportation domain by the design of a new strategic tool, which is not only geared to the specific operational requirements of PBTA but also extends conventional BMC by incorporating internal and external sustainability components.

Second, the ratings of our model by global PBTA experts from four continents indicated both convergent and divergent inclinations. The degree of convergence/divergence among experts is found to be sensitive to the hierarchical layers of the model, as the degree of divergence intensifies for specific components positioned at the third layer. Furthermore, more divergence among perceptions of experts is observed about the external components of the model compared with the internal components. The most convergent components of the model are value propositions, cost structures, and channels for the internal side, whereas the scores of experts diverged for customer segments, impact, and key partners. For external components, economic and technological factors represent the most divergent points, while there is the convergence of opinions on the importance of climate change, population dynamics, and tax policy.

Third, contrary to the claims of the convergence thesis, which underscores the significance of economic and technological factors as the main drivers of homogenization, our study shows that the perceptions about the importance of economic and technological factors diverge among global PBTA experts. Although the convergent practices of the economic and technological domain are particularly visible and are characterized more by their material aspects [34–36], the experts seem to vary across continents about the degree of their performance impacts on the public bus transportation domain. Therefore, as stated by the proponents of the divergence thesis, the perceptive understanding of material factors represents a critical factor in explaining convergent or divergent social action [37,125]. On the other hand, based on our results about the relatively more significant and more convergent scores of global experts on the internal components, and more specifically on the activity component, the over-socialized theoretical assumption of the divergence thesis, which considers social action is almost entirely shaped by local contextual factors, is challenged. The results indicate that the perceptions of experts are not only shaped by local external factors but also by their perceptions about the capacity of action that they can take. The resource repositories, existing service offers, partner networks, and governance structures of the PBTA across the globe display major differences among each other. Therefore, the perceptions of understanding and acting on the external factors are also conditioned by the perceptions of each PBTA's particular resource and activity sets. Hence, it can be argued that the divergence of opinions among the importance of model components is not altogether caused by the perceptions of local external factors but also by the perceptions of internal components as well as the capacity to link external and internal components. These results strongly support the reconciliation of the theoretical dualism between divergence and convergence of social action by focusing on specific but enduring aspects of human activities [128] such as healthcare, mobility, entertainment, politics, and economy with methodologies that take different levels of abstraction into account. Thus, the results of this study contribute to the understanding of convergent and divergent management practices in the public bus transportation domain and expand the lingering dualism of convergence and divergence by offering a particularly articulated–both in terms of inquiry domain and degree of abstraction-duality position.

Fourth, the iterative consultation with the experts of IBBG about the viability of the designed model in terms of its deployment as a strategic tool for PBTA has prompted the development of a stepwise methodology of use. Although there are existing methodologies to put BMC into use these methodologies do not particularly address how local and global external pressures can be aligned with the BMC components. Besides, with the intensity of globalization, increasing the temporal requirements of these assessments have become gradually shorter. The rapid evolution of SARS-CoV-2 disease from a regional epidemic

into a pandemic that affects the lives of billions of people is a recent example of such immediate effects. Following the short and long-run effects of global and local external pressures, which stimulate convergent or divergent practices on the part of PBTA, two distinguished strategic action steps are designed. While short-run action steps illustrate the rapid response guideline to comply immediately with rather unambiguous external forces, the long-run response guideline introduces a continuous assessment cycle to align PBTA with changes occurring in different components of the external environment. The strategic response guidelines are designed based on the agile framework, which endorses functionally diverse task forces to create minimum viable solutions in rapid iterative steps.

Consequently, the relatively little emphasis placed on the customer segments component by many PBTA across the globe indicates an important development point, as the long-run strategic action steps are triggered by gaining insight from customer segments with regard to the changes in the external model components. Exploration and commercialization of different customer segments by offerings of passenger and travel data may only become possible by intensifying the customer orientation of PBTA. Besides, PBTA need to revitalize their resources and capabilities by investing more in data-driven systems and services, as their operations, partners, and offerings will require more and more expertise in these domains. In this sense, PBTA will be more likely to transform themselves into an ICT-based multi-sided service platform, which brokers various services offered by strategic partners for different individual and enterprise customers. While many PBTA around the globe seem to understand these requirements, they still need a more systematic tool to reconfigure their resource and capability bases, making BMC an essential tool in their future operations.

**Author Contributions:** Conceptualization, B.B.; methodology, B.B.; software, B.B.; validation, B.B. and M.E.; formal analysis, M.E.; investigation, B.B.; resources, B.B.; data curation, B.B.; writing—original draft preparation, B.B.; writing—review and editing, M.E.; supervision, M.E.; project administration, M.E. All authors have read and agreed to the published version of the manuscript.

**Funding:** This research received no external funding.

**Institutional Review Board Statement:** Not applicable.

**Informed Consent Statement:** Not applicable.

**Acknowledgments:** The authors want to thank IBBG members for assessing the proposed business model framework.

**Conflicts of Interest:** The authors declare no conflict of interest.

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
