# Peer review of "Convergence or Divergence among Business Models of Public Bus Transport Authorities across the Globe: A Fuzzy Approach"

_sustainability, doi:10.3390/su131910861_

Round 1
Reviewer 1 Report
- Is it recommended for the authors to write using passive sentence/reporting style, not in active sentence. This comment applies for the whole manuscript, please revise.
- The study adopts the GUEST framework offered by Perboli and his colleagues… please put reference.
- It is recommended to merge between Introduction and Literature Review sections as one section only – 1.0 Introduction.
- 2. Surbey design framework. Change the word surbey to survey.
- Section 4.1. Discussion. Discussion should be discussed earlier in the Results and Discussion section.
- Conclusion sections should not has the subsections. Write it continuously.
- English usage needs to be improved.
Reviewer 2 Report
The topic taking into account in the paper is interesting, but unfortunately, the body of the paper does not bring anything new. First of all, the Authors should correct the following issues:
- the paper should be formatted according to the Sustainability template requirements,
- the "Fig. 1. Business model framework for public transportation authority regarding P.E.S.T.E.L. analysis." There is a table or figure?
- Is "Fig. 2. Surbey design framework" should be "Fig. 2. Survey design framework" - typos,
- The process of the survey presented in figure 2 is obvious and well-known,
- the description of multi-criteria-decision-making methods is also redundant as these are commonly known methods,
- the Authors wrote "Abbreviations of the main and sub-criteria are presented in the Appendix." The paper is not long, so in order to make the paper more readable, the abbreviations should be placed near/below table 2, where they are used,
- in Table 2, we can find the items without explanations, e.g. "*",
- concern the subsection "4.1. Discussions, limitations, and future work" in the Conclusion section. The discussion of the obtained results should be included in the paper text before the Conclusion section.
In conclusion, the paper in its present form has major deficiencies. The Authors should prepare the paper with greater care and re-submit it for review.
Round 2
Reviewer 2 Report
The Authors improved the majority of reviewer comments.
Thank you very much.